# Exploring Partners’ Part in Shaping the Home Food Environment During the Transition to Fatherhood

**DOI:** 10.3390/nu16244356

**Published:** 2024-12-17

**Authors:** Chagit Peles, Mary C. J. Rudolf, Danielle Shloim, Netalie Shloim

**Affiliations:** 1Azrieli Faculty of Medicine, Bar-Ilan University, Safed 1311502, Israel; mary.rudolf@biu.ac.il; 2School of Education, Language and Psychology, York St John University, York YO31 7EX, UK; danielle.shloim@yorksj.ac.uk; 3School of Healthcare, University of Leeds, Leeds LS2 9JT, UK; n.shloim@leeds.ac.uk

**Keywords:** expectant fathers, first pregnancy, human-nesting, prenatal nutrition, home food environment

## Abstract

Objective: To investigate primiparous women’s partners for knowledge, attitudes, and practices regarding the physical home food environment (PHFE), and to assess if the first pregnancy provides a teachable opportunity to enhance the PHFE of first-time pregnant couples. Design: This was a two-phase longitudinal in-depth qualitative study involving questionnaires and individual interviews during and after pregnancy. Participants: Fifteen male partners of primigravida women. Main outcome measures: Knowledge, attitudes, and behaviors concerning PHFE; lifestyle and dietary habits; and interest in guidance regarding healthy PHFE during the first pregnancy and the transition to parenthood. Analysis involved descriptive statistics and thematic analysis for qualitative data. Results: Key findings include the importance of mutual prenatal PHFE decisions; increased motivation for a healthier PHFE during pregnancy; a desire to provide a healthier environment than their own parents offered, acknowledgement of their wives as ‘nutrition experts’; challenges in accessibility of health foods in the home; and ‘the child eats what we eat’. The first pregnancy was recognized as a critical period for establishing a healthy PHFE. Conclusions and Implications: Expectant fathers have a crucial role in nestrition (nutritional nesting) in first pregnancies. Their engagement is essential in establishing a more supportive nutritional environment in the home and influencing the family nutrition in the long term. There is a need to promote partner involvement, investigate the broader roles of expectant fathers and non-male partners, and develop effective PHFE education for couples in the first pregnancy.

## 1. Introduction

First pregnancy is a crucial transitional period for expectant parents, presenting an opportunity for nutrition education [1]. While often joyful, significant tensions may occur due to the dynamic changes involved [2,3]. During this time, expectants are highly engaged with healthcare services and are receptive to health messages, making pregnancy-specific nutrition information an opportunity to safeguard babies’ health [4,5].

Prenatal conditions at critical developmental periods can have lasting effects, potentially predisposing offspring to obesity [6]. This highlights the need to address modifiable prenatal factors as targets for early nutritional interventions [7]. The physical home food environment (PHFE) is one such modifiable factor influencing food intake and the development of obesity in early childhood [8]. As the transition to parenthood impacts dietary habits and food choices [9], a critical opportunity presents for the adoption of healthier nutritional practices in the home. However, research on dietary changes during this transition is mixed [10,11]. While there is evidence that positive changes may occur due to more structured family meals and healthier food purchases, the arrival of a child can also lead to an increased consumption of processed and fast foods high in fats and sugars [12,13,14].

Assessing the knowledge, attitudes, and practices of first-time pregnant and postpartum parents is therefore important to identify specific factors influencing how the transition to parenthood affects dietary patterns and the home food environment. It is particularly important to examine these factors separately for both partners, as research shows that eating behaviors can differ significantly between fathers and mothers [15]. However, to date, few studies have focused on men’s experiences on transitioning to parenthood, particularly regarding health and nutrition. A survey of the literature reveals little emphasis on changes in paternal eating behaviors, especially during and after the first pregnancy [14,16,17].

Research on the physical home food environment during this transition also often overlooks fathers [9,18], highlighting the need for comprehensive studies exploring how both parents influence the home food environment [19]. Investigating ways to prepare for a healthy PHFE among first-time pregnant families is essential, given parents’ roles as gatekeepers and models [11], significantly impacting their children’s dietary habits and health over time as well as potentially contributing to the worrying and increasing prevalence of obesity and non-communicable diseases over the life course.

This study is the second component of a research program that embraces a new concept which encompasses *nesting* in the matter of nutrition: **nestrition** or nutritional nesting. In a series of studies it explores how expectant parents might be guided during their first pregnancy in the creation of a healthy home food environment. The first paper [20] explored pregnant women’s perspectives and the potential of first pregnancy as a teachable moment; this phase focuses on the experiences of their partners.

In a similar way to the first **nestrition** paper, the aims were twofold. First, to examine the knowledge, attitudes, and practices of first-time expectant and postpartum fathers pertaining to the PHFE, and to compare the results with the mothers’ results [20]. Second, to explore if they viewed a first pregnancy as a teachable opportunity to enhance the couples’ PHFE. Understanding the intricacies of the PHFE during the transition to parenthood is pivotal prior to developing targeted interventions that promote healthier dietary practices. By focusing on both parents, especially partners’ perspectives, we aimed to explore how a supportive, health-conscious home environment can be better established, benefiting long-term family lifestyle and health.

## 2. Methods

### 2.1. Study Design

This two-phase longitudinal qualitative study, conducted between 2020 and 2022, focused on the male partners of first-time pregnant women who had participated in the nestrition study [20] (hereafter termed expectant fathers). The methodology replicates the nestrition study and comprised two phases: initially, primigravida expectant fathers completed an online questionnaire to assess their attitudes and practices related to their PHFE. Subsequently, individual face-to-face interviews were conducted by the first author (CP) at the participants’ homes, each lasting approximately one hour. Postpartum, the same participants completed a follow-up online questionnaire and were re-interviewed to evaluate changes since childbirth. The second phase aimed to explore updated perspectives and practices regarding their PHFE during the transition to fatherhood. The COREQ checklist (Appendix A) was used as a guide for reporting the design, data collection, analysis, and findings.

### 2.2. Recruitment and Participants

First time expectant couples were recruited via advertisements on social media groups for expectant parents, their dietitians and other pregnancy-related websites. Additionally, printed flyers were distributed at the host university, local libraries, and conferences at the local hospital. Inclusion criteria for this study included being the partner of a first-time pregnant female participant in the nestrition study and being over 18 years old.

### 2.3. Study Materials

#### 2.3.1. Semi-Structured Interviews with *n* = 15 in Pregnancy and *n* = 12 Post Pregnancy

Replicating Peles et al. [20], semi-structured face-to-face interviews were conducted with the expectant fathers during their partner’s pregnancy and again at least six months post fatherhood. These interviews adhered to a guide developed by the authors, who have extensive clinical and research expertise in pregnancy nutrition (see Appendix A). The guide facilitated discussions on relevant issues within a structured framework to understand participants’ attitudes, perceptions, and motivations [21]. Topics included first pregnancy experiences, perceived and actual PHFE, barriers and facilitators to maintaining a healthy PHFE, the potential of pregnancy as a ‘teachable moment’ for establishing a healthy PHFE, and opinions on PHFE classes. Each interview began with a brief informal check on the interviewee’s well-being and, in the second interview, the baby’s well-being. The same guide was used for both time points. Data collection continued until thematic saturation was reached [22], defined as the point where no new themes emerged. Interviews were conducted and recorded by the first author, a clinical dietitian with 25 years of experience working with parents of preschool children (Appendix A). Transcriptions were performed by an independent individual not involved in the research.

#### 2.3.2. Questionnaires with *n* = 13 in Pregnancy and *n* = 14 Post Pregnancy

The questionnaire (Appendix A) was administered online before the interviews in both study phases. Derived from validated instruments addressing various PHFE components [23,24], it comprised 31 items and took approximately 20 min to complete. Questions covered food preparation practices, a food inventory relevant to childhood obesity (including fruits, vegetables, whole grains, high-fat foods, snacks, and beverages), the partner’s perceptions of PHFE, food shopping habits, breastfeeding intentions, and interest in nutrition programs. The Food Frequency Questionnaire (short version of the validated Hammond FFQ [25,26]) included foods related to risk and protective factors for childhood obesity in pregnant women’s diets [27].

Sociodemographic questions included educational attainment, classified on a scale ranging from ‘No formal schooling’ to ‘Postgraduate degree’, household income, and medical problems. Household food insecurity was assessed through two self-reported measures: (1) “The food that we bought just didn’t last, and we didn’t have money to get more.” and (2) “We worried whether our food would run out before we got money to buy more.” [28]. Body mass index (BMI) was calculated at two different time points based on self-reported height and weight measurements. Further details are presented in Table 1.

### 2.4. Data Analysis

The analysis followed Braun and Clarke’s (2006) method for thematic analysis [29], a foundational method for qualitative data analysis that allows for the flexible identification and categorization of patterns in experiences or meanings, as described in the previous nestrition study [20]. The analysis was inductive and conducted at a latent level, aiming to uncover underlying meanings, assumptions, and patterns in the participants’ accounts. The term ‘expectants’ was used for participants during the pregnancy and ‘fathers’ for postpartum. The second set of interviews, conducted postpartum, assessed how participants’ perceptions and practices concerning PHFE changed after childbirth, allowing us to examine both continuity and change as they transitioned from pregnancy to fatherhood.

The methodological approach included the following:

*1 Familiarization with the data*: The first author, a bilingual researcher, translated all interviews into English and annotated the transcripts to identify prominent themes or meanings. The senior author, an expert in qualitative research, reviewed a randomly selected subsample of six interviews at each time point (12 in total) to ensure reliability and depth of analysis. This subsample size, deemed sufficient for thematic saturation according to Guest et al. [30] was reviewed until full agreement was reached.

*2 Coding*: Descriptive labels or codes were assigned to text segments that captured the meaning of the data. This process was conducted transcript-by-transcript and line-by-line, grouping related text under common codes. Codes were iteratively refined and expanded to incorporate all relevant data. The first author completed the coding for the entire dataset, while the senior author repeated the process for a subsample of interviews.

*3 Generating Themes*: Commonalities between codes were identified, and these similarities were clustered to create preliminary themes. The first author produced these themes, which were refined through discussions with the other authors. At this stage, all data were reviewed and agreed upon by all authors.

*4 Reviewing the Themes*: The themes’ validity was evaluated by examining their coherence with the data and refining them based on new insights. This review ensured the themes accurately reflected the data.

*5 Defining and Naming Themes*: The final set of themes was named, and a description of their core characteristics was written. The analytic output was discussed and confirmed by all authors to ensure it was grounded in the data.

This process allowed us to thoroughly explore and understand the shifts in men’s experiences as they moved from pregnancy to postpartum, providing insight into the challenges and opportunities at each stage.

### 2.5. Quantitative Analysis

Participant characteristics were summarized via the median and IQR, and the FFQs were summarized by the mean and SD. The FFQ analysis concentrated on fruit, vegetable, whole grain, and legume consumption as markers for a healthy diet, while high-fat processed foods, sugar-dense items, processed meat products, crisps, and sugar-sweetened beverages were examined as indicators of a less favorable dietary profile. Given the sample size, inferential statistical testing was precluded when comparing measurements obtained at the two timepoints under study.

### 2.6. Ethical Approval

This study received full ethical approval from the Bar-Ilan University Faculty of Medicine Ethics Committee (Approval No. IRB09-2019). Written informed consent was obtained from all participants.

## 3. Results

### 3.1. Participants’ Characteristics

The 15 participants were all male, married, had no previous children, and had completed high school, with eight holding a bachelor’s or master’s degree; their ages ranged from 24 to 32 years (M = 27.7, SD = 2.5). All were healthy; four reported BMI scores > 25, ranging from 26 to 41 kg/m^2^. The first phase took place in mid to late pregnancy and the second when the participants’ babies, all born full-term, were between 6 and 13.2 months old (M = 10, SD = 2.7). During pregnancy, all were interviewed and 13 completed the questionnaire. In the fatherhood follow-up, 12 were interviewed and 14 answered the questionnaire. Terminologically the term ‘partners’ refers to the male participants during pregnancy, while ‘fathers’ denotes the participants in the postpartum phase. Table 1 presents participants’ characteristics.

**Table 1 nutrients-16-04356-t001:** Participants’ characteristics—longitudinal study (*n* = 15).

Participant	During Pregnancy							Post Pregnancy				
	Edu ^1^	Age (Year)	Income ^2^ (USD/month)	GA (Weeks)	BMI (kg/m^2^)	Healthy Lifestyle ^3^ (1–10)	Health Satisfaction ^4^ (1–5)	Income ^2^	BMI (kg/m^2^)	Baby’s Age (Month) *	Healthy Lifestyle ^3^ (1–10)	Health Satisfaction ^4^ (1–5)
P.A	5	24	-	13	20.8	10	4	-	21	6	9	4
P.B	7	30	-	14	24.3	8	4	4000 USD/m	24	13	7	4
P.C	6	30	4000 USD/m	20	24.3	9	4	4000 USD/m	24	7	8	2
P.D	6	27	670 USD/m	24	22.1	5	4	1400 USD/m	23.5	8	8	3
P.E	5	26	2000 USD/m	23	20.1	7	5	2000 USD/m	*23*	11	5	5
P.F	5	27	4000 USD/m	26	27.4	6	2	2000 USD/m	28.4	12	8	4
P.G	6	24	2000 USD/m	27	22.2	8	4	2000 USD/m	22.5	7	7	4
P.H	5	27	1400 USD/m	29	30.1	5	3	1400 USD/m	30	8	7	3
P.I	5	31	2000 USD/m	31	41.5	10	3	-	-	-	-	-
P.J	6	26	-	32	18.3	5	3	-	18	7	8	4
P.K	5	32	-	33	-	-	-	4000 USD/m	28	12	7	4
P.L	6	25	2000 USD/m	32	19.3	7	2	4000 USD/m	19	11	6	3
P.M	7	30	-	37	-	-	-	4000 USD/m	21	12	9	4
P.N	6	28	2000 USD/m	37	21.7	8	4	2000 USD/m	22	12	9	4
P.O	5	28	4000 USD/m	37	26.2	7	5	4000 USD/m	27	13	7	4
Mean (SD)	NA	27.7 (2.5)		27.8 (7.7)	24.5 (6.1)	7.3 (1.8)	3.6 (0.9)		23.6 (3.6)	10 (2.7)	7.5 (1.2)	3.7 (0.7)
Median	NA	27		29	22.2	7	4		23.2	11	7.5	4
IQR	NA	26–30		23.5–32.5	20.8–26.2	6–8	3–4		21.2–26.2	7–12	7–8	3.3–4

**^1^** “What is the highest level of education you completed?”: 5 = high school; 6 = college/university; and 7 = postgraduate. **^2^** “What is your monthly household income before tax and including benefits (converted to $): 6 = prefer not to say. **^3^** “Overall, how healthy do you think your lifestyle is at the moment?”: 1 = not at all to 10 = very healthy [31]. **^4^** “How satisfied are you with your health?”: 1 very dissatisfied to 5 very satisfied [32]. * Baby’s age in months at the time of the questionnaire.

### 3.2. Questionnaire Analysis

#### FFQ (Food Frequency Consumption)

During pregnancy, participants averaged 4.4 daily servings of fruits and vegetables, with six of thirteen meeting the 5+ servings recommendation (see Table 2). During fatherhood, a decline in fruit and vegetable consumption was noted, with the decrease predominantly in vegetable intake. The accessibility of fruits and vegetables in the home decreased during parenthood, reflecting similar findings. The participants’ consumption of coffee and low-calorie drinks during pregnancy remained stable post pregnancy, whereas sweet beverage consumption decreased in fatherhood. The reported consumption of high-fat processed foods, high-sugar foods, and savory snacks remained consistent pre- and postpartum (Table 2).

### 3.3. Physical Home Food Environment (PHFE)

The analysis of the Home Food Environment data revealed that certain vegetables were consistently available during both pregnancy and parenting. Onions were present in all households, while cucumbers, tomatoes, carrots, potatoes, apples, and bananas were in 14 homes. During parenthood, the participants indicated no hard candies in the home (with one father noting they are reserved for guests and children). However, the availability of sweetened drinks increased, while ‘100% fruit juice or fruit and vegetable combination’ decreased.

### 3.4. Thematic Analysis

Five themes relating to the PHFE emerged from coding the prenatal and postpartum interviews among the partners: the importance of prenatal decisions on the home food environment, aiming to bridge differences between partners; wives’ interest in dietary guidance peaks during pregnancy, aspiring to provide a better home food environment for their child than they had had themselves; a recognition of their wives as the primary nutritional expert; encountering challenges in accessing healthy foods at home; and post pregnancy, an emphasis on feeding their children what they, as parents, consumed, depending on available household food and nutritional challenges. These themes are described in Table 3.

### 3.5. Theme A: The Importance of Mutual Prenatal PHFE Decisions

The first theme focuses on the significance of mutual decisions, such as the selection of food for the home prior to the baby’s birth, in order to improve nutrition and bridge differences. This theme, which reflects the complexity of nutrition as a domain where partners bring differing habits from their own childhood homes, can lead to tension if not openly discussed.

In constructing the shared home, the food environment is created by both partners influenced by their own past, consciously or unconsciously. Based on the interviews, it seems that while some partners did not maintain healthy eating habits, their wives valued healthy eating practices. Despite differences and initial difficulties, it was evident that common ground could be found with the adoption of healthier lifestyle habits, especially during pregnancy.

“… I come from a family background that doesn’t pay attention to nutrition—for example, there are sweetened beverages and sweets…. Since we married, I have “weaned off”, thanks to A. …. It started even before pregnancy and now it is twice as much… it was difficult for me at first, but now I’m fine… we got used to it (laughs)…… I guess her family is probably healthier relative to my family…” (P.B; during pregnancy).

In light of differences, other interviewees described adopting a bridging strategy of respectful and egalitarian discourse, which they considered a positive step in terms of health—a push toward a “better place”:

Interviewer: “As you mentioned, H. is more concerned with nutrition and has a higher awareness. Is that okay with you?”

P: “It’s really good. I love it. I love that H. pushes me to better places. I don’t mind. And it’s not oppressive, she does it gently. We allow it, I can bring whatever I want to this house, but I am less comfortable bringing it” (P.H; during pregnancy).

Difficulties tended to manifest more prominently when prior perceptions and habits from childhood and adulthood were not openly discussed. When this occurred, particularly during pregnancy, relationship tension and conflict could arise.

“It’s an issue that is literally on the table. I see nutrition and fitness as important—daily protein, fruits, and vegetables. C. is neglectful and unaware of this. The issue of sharing cooking has become more significant in our marriage lately, especially as she has less physical strength and wants to educate me.” (P.I; during pregnancy).

Overall, the findings highlight the importance of reciprocity, suggesting that respectful dialogue can promote a healthier home food environment and a potential for positive dietary change.

### 3.6. Theme B: Increased Motivation for a Healthier PHFE During Pregnancy

The second theme indicates that expectant partners are more motivated during pregnancy to create a healthier physical home food environment (PHFE) than after birth. They viewed pregnancy as a “window of change” with significant nutritional implications, recognizing the importance of caring for their pregnant wives and making health-promoting decisions.

“I take care of her and her diet. She is pregnant, so I make sure that she eats things that will be helpful for her and our baby… We ask ourselves every day what we did today for the benefit of the pregnancy?” (P.A; during pregnancy).

Several participants specifically pointed out the challenges associated with pregnancy-related symptoms and the noticeable differences in experiences between genders. They emphasized the critical role of the non-pregnant partner in fostering a supportive and healthier PHFE during this demanding period of pregnancy. This support is seen as essential not only for the physical well-being of the pregnant partner but also as a fundamental aspect of nurturing a supportive family environment during these transformative months.


*“There’s a gender difference when it comes to diet during pregnancy. The woman go through so many changes, it can be tough to switch up her eating habits… For us men, it’s a bit easier since we don’t have those physical changes holding us back. It really comes down to how much we want to step up and make our home food environment better, it is a question of desire and family value…” (P.J; during pregnancy).*


An important finding was that interviewees emphasized that by the time they became parents, their eating habits had already been shaped and were difficult to change. Creating a healthier food environment, was therefore deemed more practical and feasible:

“Pregnancy itself… to influence the fetus seems to me to be something that can work… We hope it will continue post-pregnancy…. For the majority, pregnancy happens at a stage when your nutritional patterns and behavior have already been shaped… It’s usually very difficult to change your eating habits, but the food environment is easier to change at our age…” (P.M; during pregnancy).

“In the end, it is a matter of adopting habits and then taking advantage of pregnancy momentum, taking advantage of the entire nine months with all its excitement. As a result of the first pregnancy, there is a great desire: from now on, we eat healthy…. But changing habits is difficult, so at least changing what is on the table will help….” (P.G; during pregnancy).

In the second phase of the study, most interviewees continued to emphasize the importance of the physical food environment at home. They expressed that this stems from the desire to give their firstborn child the best possible healthy start. A healthy home food environment helped them to build nutritional habits with their child and made it easier to implement a healthy lifestyle for their infant.

Nutrition at home should be appropriate from the start. You have to build an environment from the moment a child is born. It is possible that without noticing he will eat whole grain bread and get used to it. It is not easy to change your lifestyle because it is complicated to implement…. (P.L; in fatherhood).

On being asked about classes/workshops to help couples foster a healthier lifestyle for the new family, there was consensus for short, practical meetings that provided information concerning the “here and now”, rather than long, detailed nutrition workshops:

“Workshops for a healthy lifestyle—I don’t particularly like these things. I know what’s best for me and what’s not. I choose my way.…. I don’t need someone to tell me that. I need someone to guide us on how to make healthy and fast snacks….that’s worthwhile.” (P.O; in fatherhood).

Data from the questionnaire supported the qualitative findings. During pregnancy, partners expressed more interest in nutrition workshops than during fatherhood, and they were also less interested overall than their wives.

#### B.1. Voicing a Preference for a Healthier Home Environment than They Had Experienced in Childhood

A sub-theme concerning high motivation during pregnancy relates to the desire to establish a healthier nutritional environment for their unborn child than what participants themselves experienced during childhood.

“I feel ignorant about nutrition. I wish to improve from where I am right now… and I am afraid of instilling the sins of my parents in our child.” (P.F; in fatherhood).

According to the interviewees, children are born into a nutritional reality and environment set by their parents from birth. Growing up, nutritional decisions are made within the limitations of the nutritional environment formed at home, as expressed in the following:

“…We didn’t have such a healthy home… When I was at my parents’ house, I ate what was available… Since health at home is dictated, there isn’t much left to think about… There’s nothing you can do, you live there… Now, when I build a house with R., and particularly during pregnancy, I want my child to eat healthier than I did as a child” (P.A; during pregnancy).

Some interviewees noted that their childhood PHFE had improved over the years. Whether this was due to their parents’ health problems as they aged, or was due to greater parental availability and awareness as children grew is unclear. Some expressed a strong motivation to improve family dietary habits they had inherited from their childhood. For instance, the following:

“We eat family meals both of us in front of the television on the sofa. Having grown up in my parents’ household, I have developed this bad habit… In my family, we rarely had family meals. We intend to eat around the table and spend less time staring at the television after the birth. I believe that it should be that way instead of ‘individuals sneaking around’ like we did when I was growing up”. (P.N; during pregnancy).

### 3.7. Theme C: Acknowledgement of Their Wives as ‘Nutrition Experts’

The third theme focuses on partners recognizing their wives’ role in nutritional decisions. Most reported that they were less involved in both nutrition and the PHFE than their wives whom they saw as experts in these matters. Some described themselves as serving more as the “executive arm” for implementing decisions. For example, P.E explains as follows:

“We have a book that compares the home to the government, so S is the food authority. Generally, I only buy and eat (laughs)… I’m less concerned with managing nutrition at home. It comes from S. How do I choose what to buy at the supermarket?—I have a very organized list from S.” (P.E; during pregnancy).

The practical application of their wives’ knowledge was clear in meal preparation and planning. Wives typically determined the menu while husbands handled shopping. The questionnaire results also showed women were usually the primary cooks. A third of couples found this arrangement convenient, relying on their wives as experts while remaining passive.

The perception of wives as ‘Nutrition Experts’ seemed to stem from their greater knowledge and interest in health and nutrition in-utero and post-birth within the domestic family space. This might coincide with the wives being more involved and active in these matters and may have encouraged partners to assume a more passive role.

### 3.8. Theme D: Challenges in Accessibility of Healthy Foods in the Home

The fourth theme highlights a gap between the availability and accessibility of healthy foods, particularly raw vegetables, cooked meals, and healthy snacks. Interviewees emphasized that simply having healthy foods available at home is only the first step; access is critical for consumption and depends on preparation time, cooking skills, and proper planning. This challenge is especially pronounced for fresh fruits and vegetables, which have a limited shelf life and spoil quickly, as P.D described during pregnancy and as a young father: *“Healthy eating requires making healthy foods both available and accessible. Now I eat vegetables almost every day. If A. cooks, I eat as well. Since we got married, I eat a lot more vegetables because they are readily available” (P.D; during pregnancy)*

“There is often a shortage of food at home before shopping. We must plan our shopping correctly… when to buy, how much, and how to store it… not just what to buy, in order for it to be available throughout the week.” (P.D; in fatherhood).

Participants noted that both free time and the ability to cook healthy meals decrease after giving birth due to the often exhausting demands of caring for their newborn. As fathers, they identified several practical challenges such as the need for readily available fruit in the kitchen, washed, cut vegetables in the refrigerator, and cooked vegetables, legumes, and complete meals prepared for storage. Pre-prepared meals and healthy snacks were deemed essential. P.J exemplifies this in the following:

“Our meals are prepared once every three days, we plan ahead, and then we take what we need. It is essential for me that I have food prepared in advance to quickly take to work in the morning…It is important to have fruits and vegetables accessible and easy to prepare without much effort. ……. This is important, but currently difficult to implement.” (P.J; in fatherhood).

Ten partners reported relying on their wives for cooking, making it less likely that healthy food would be accessible at home during late pregnancy and early parenthood due to exhaustion and preoccupation with childcare. This led to a discrepancy between the intention to consume healthy food and its availability. As Shai pointed out, the gap in healthy food accessibility tends to widen after childbirth if not properly addressed.

“I dream of how our refrigerator should be filled. Why a dream? Because I don’t cook—I don’t know how to cook. I like fruits, but I don’t have the strength to cut them…. I have no problem having a salad every day, but I don’t have the energy to prepare it. On the other hand, if C. brings me a ready-made good salad, I eat it with great pleasure.” (P.C; during pregnancy).

“We depend on my wife to cook since I do not prepare meals. C. is much less capable of preparing our dinners now, so I eat less healthily as a result. It really bothers me that there is no accessible healthy food at home.” (P.C; in fatherhood).

### 3.9. Theme E: “The Child Eats What We Eat”

Fathers emphasized that their firstborn eats the same food as they do, as the child becomes an integral part of the newly formed family. They explained that their baby ate from the accessible options available at home, which helps integrate the child into the family’s eating habits. They highlighted that the home food environment that existed during pregnancy continues to influence their food choices in the early stages of parenthood. Jonathan explains this in the following:

“She is still nursing a little and also eats what we eat. When we started to introduce her to solids, she only ate a little, but she had no problem with anything. She agreed to taste everything. Now she eats much more in terms of quantity and eats everything. We give her what we eat, except for honey and things we were told not to give her yet at this age.” (P.M; in fatherhood).

Most partners wanted to maintain a healthy home food environment as their infant began complementary feeding. They discussed improving the food environment through thoughtful planning around food procurement, cooking, and storage, ensuring variety, choosing healthier yet potentially more expensive options, avoiding junk food, cooking healthily, and minimizing empty carbohydrate consumption. *“We want to be healthier now in preparation for the birth and, when the baby is born, we will increase the variety of healthy things. We would like her to be familiar with healthy options.” (P.B; during pregnancy).*

During the transition to parenthood, fathers reported increased motivation to eat healthily to serve as role models for their children. They aimed to provide a healthier start than they had, exposing their kids to a variety of healthy foods. This approach helped babies become accustomed to healthy tastes and encouraged them to embrace nutritious options. However, some fathers acknowledged that time and energy constraints often lead to eating processed foods high in fat and sugar due to their convenience. One father shared a unique approach, stating, “We Eat What the Child Eats”, emphasizing that he and his wife plan to improve their nutrition by participating in the tasting phase with their child and trying new foods together.

## 4. Discussion

This study aimed to examine the knowledge, attitudes, and practices of first-time pregnant and postpartum male partners regarding the physical home food environment, compare findings with their pregnant spouses from the nestrition [1] study and explore whether a first pregnancy offers a teachable moment to improve the PHFE. Our research adds to the literature by focusing on male partners’ perspectives during the transition to parenthood, an area that has been underexplored. The findings provide valuable insights into the role of expectant fathers in shaping the PHFE, highlighting their significant influence during this critical period.

Expectant fathers showed greater motivation and interest in healthy nutrition during pregnancy than they did during fatherhood, viewing pregnancy as a “window of change” with significant nutritional implications for the fetus and family. They recognized the importance of supporting their pregnant wives’ health, often adjusting their nutrition during the first pregnancy. This aligns with Versele et al.‘s findings [15] that men’s health motivation during pregnancy is driven by a desire to support their partner and future child. Our results highlight that expectant fathers play a crucial role in addressing their wives’ nutritional needs and cravings. Unlike pregnant women, men do not undergo physical changes or altered taste senses, making their support vital. The involvement is particularly prominent during pregnancy, childbirth, and the early months of parenthood. Our findings emphasize that the first pregnancy is a critical period for expectant fathers, presenting an opportunity for targeted interventions to promote sustained dietary changes within the new family. Participants emphasized the importance and feasibility of establishing a healthy physical home food environment during pregnancy, motivated by their age, the pregnancy itself, and the anticipation of their first baby. Acknowledging the challenges of changing entrenched eating habits, they viewed focusing on creating a healthy food environment as a practical step. The first pregnancy was seen as an ideal opportunity to cultivate a desirable nutritional setting for the family.

Once they became fathers, the significance of the PHFE was underscored as they aimed to give their firstborn a healthy start and instill good nutritional habits. However, stress and decreased mental well-being during early fatherhood often leads to a decline in nutritional quality [34]. Versele et al. [15] identified changes during the transition to parenthood, such as time constraints and food availability, which act as both barriers and facilitators to a healthy home food environment. Both women and men reported difficulties with self-control, such as succumbing more quickly to sweets, during pregnancy and postpartum. Exclusively, fathers mentioned instances of self-licensing in the postpartum period, primarily to justify increased indulgence, like consuming more comfort foods, due to fatigue or a lack of motivation to cook. This highlights the need for expectant fathers to establish a healthy PHFE early to mitigate dietary quality decline.

We found that increased motivation occurred around the weaning stage, where fathers made greater efforts to set a personal example by introducing nutritious foods. Research indicates that transitioning to fatherhood can lead to nutritional improvements, such as decreased alcohol consumption, but it can also heighten the risk of weight gain and declining self-rated health [14,17]. These insights underscore the necessity for targeted research and interventions to support partners during their transition to first-time parenthood.

Partners in our study aimed to create a better nutritional environment for their children than they had experienced in childhood. They noted that children inherit their parents’ nutritional framework from birth, which influenced their dietary choices. This tendency to compare their current and early home food environments emerged among the men but was not found with their spouses [20]. The generational comparison may reflect current sociological factors, where young fathers assume different roles than their fathers [35], leading to a generational comparison in nutritional matters, which is a significant part of the responsibilities of new parents. Research shows that men, particularly in early fatherhood, are more inclined toward social comparison and competitiveness to measure success and gain acknowledgment. In societies with distinct gender roles, men tend to be more competitive while women are more collaborative, with fathers feeling a strong need to excel to meet societal and familial expectations [36].

The participants emphasized the need for healthy food to be accessible at home, highlighting the disparity between availability and accessibility. This ‘accessibility gap’ significantly contributes to the low consumption of healthy food, as unhealthy options are generally more accessible. The finding aligns with key models in the field, such as the Model of Home Food Environment Pertaining to Childhood Obesity [8] and NCCOR’s model [37], which emphasize that both the availability and accessibility of food at home are crucial in preventing childhood obesity. Interestingly, the two aspects of the physical home food environment—basic healthy food availability and, in addition, the need for these foods to be accessible—emerged from the combined data of both parents in our research group.

A key reason for partners’ emphasis on food accessibility at home relates to their perception of wives as “nutrition experts”, along with the fact that women were typically the main cooks. This likely influenced men’s concern about food accessibility, as they often assumed passive roles in meal planning and preparation, relying on their wives to ensure food was available. This dynamic highlights how intra-couple roles shape the home food environment, with significant implications for dietary practices and nutritional outcomes. Other research shows how women, often viewed as primary influencers of nutrition, tend to shape meal choices, grocery shopping habits, and family health behaviors [38].

Traditional gender roles also often place women in charge of meal planning, further influencing household dietary preferences [39]. These findings align with Wirsching et al., who identified the “Health Expert” role within couples that shapes dietary practices [40]. In a similar way, our study shows that men’s reliance on their wives for food accessibility underscores the impact of intra-couple dynamics on dietary behavior. This reinforces the need to consider both partners’ roles in nutritional interventions, emphasizing the importance of addressing food accessibility alongside availability to foster healthy eating habits. Our findings that women served as the primary “nutrition experts”, echoes previous research on gender disparities in dietary influence [41]. This reflects traditional roles where women manage food preparation and household tasks, often acting as dietary gatekeepers due to their higher health literacy, positively affecting men’s diets [42]. Adding to these studies, our research, which included couples at the start of their shared journey before childbirth, found that men demonstrated motivation to engage in joint decision-making regarding the physical home food environment and beyond.

Within the marital context, interviewees emphasized the importance of coordinating food choices at home during the first pregnancy, highlighting this as crucial for establishing a healthy food environment at a pivotal time. They discussed the complexities and difficulties encountered during this process and shared strategies for bridging gaps and reaching agreements on maintaining a nutritious home environment. These insights resonate with Pauly et al. [43], who emphasized the importance of relationship transitions and concordance between partners. Their study suggests that changes in relationship status, such as marriage or cohabitation, significantly influence health behaviors and alignment between partners. Our research concludes that this concordance is particularly evident during a major life transition like first pregnancy, where joint decision-making and open communication about health and nutrition are essential. By addressing and managing their differences, couples can create a supportive and health-promoting home environment, ultimately benefiting the overall relationship and the health of the new family.

Our findings also highlight the importance of shared decision-making in shaping the home food environment for expecting parents. This adds to Versele et al.’s [13] work on the critical role of partner support and lifestyle change during pregnancy and the work of Baer et al. [39] work on the importance of intra-couple dynamics in dietary preferences (outside of pregnancy). The findings underline the complexity of dietary behavior during significant life changes and reinforce the need for couple-based strategies to foster a health-promoting environment during the transition to parenthood.

The comparison of FFQ consumption data between the partners and that of the women [20] revealed some differences. Firstly, men appeared to consume fewer vegetables and fruits overall compared to their wives at both time points. Secondly, fathers and mothers ate fewer raw vegetables and fruit during parenthood, reinforcing the challenge parents, especially fathers, face in meeting the recommended intake of fruits and vegetables during this transition. This change in dietary consumption may indicate a possible influence of the pregnancy experience on health behaviors, which seems to diminish once the pregnancy period is over. Thirdly, during pregnancy, unlike their expectant partners, men did not reduce their intake of coffee and diet drinks. This may indicate higher nutritional awareness among pregnant women to follow guidelines that recommend reducing coffee and low-calorie sweeteners for the health of the fetus. Lastly, during parenthood, mothers appeared to reduce their consumption of high-fat processed foods, high-sugar foods, and savory snacks, a change not observed in fathers. These findings suggest distinct dietary behaviors between men and women during the transition to parenthood, underscoring the need for targeted nutritional interventions that address these differences.

Our study was conducted by researchers with expertise in nutrition, medicine, psychology, and qualitative research. This allowed us to comprehensively investigate and highlight the previously unexplored role of male partners in shaping the physical home food environment during the transition to parenthood, marking a unique and novel contribution to the field. There are some limitations as well as strengths to our study. Firstly, the method of recruitment may well have attracted couples whose motivation was greater than those who did not reply. The research was also carried out in a single country. While this ensures cultural relevance for this population, it may limit generalizability to other populations. The sample predominantly included middle-class men, with limited representation from lower socioeconomic backgrounds and no evidence of food insecurity. Most of the participants were of healthy weight, so including perspectives from men dealing with overweight would provide valuable depth in future research. Lastly, the research was conducted in the later stages of the global COVID-19 pandemic, and it is important to note that both rounds of interviews were conducted after restrictions on movement had been lifted in the country, during a period when public life, supported by widespread vaccination efforts, had largely resumed.

## 5. Conclusions

The findings of our study highlight the crucial role of partners in creating a healthy physical home food environment during the first pregnancy and transition to parenthood. Rather than being peripheral, the partner’s involvement is central for several reasons. Firstly, our results indicate a heightened motivation during the transition to fatherhood alongside a role as the primary food purchaser, especially during the challenging period of pregnancy. Partners noted that unlike their pregnant partners, they did not experience physical changes or altered taste preferences which can influence the PHFE. Both the expectant mothers and their partners regarded men as important supporters of the pregnant mother’s nutritional needs and saw their involvement as crucial during both the pregnancy and the initial months of parenthood. It was evident that the shift to parenthood plays a particularly significant role in shaping health behaviors, underscoring the partner’s vital role at this time. Establishing open and effective communication and aligning the couple’s actions therefore are critical.

A further key finding relates to the accessibility of healthy food at home. Partners stressed that simply having healthy food available is insufficient; it must also be easily consumable. The gap between the ‘eye’ and the ‘mouth’—in other words, the effort required to prepare and eat the food—plays a significant role in maintaining a healthy home food environment at this time.

Our research highlights the importance of including partners in future studies and interventions, particularly in the context of nestrition (nutritional nesting) [20]. By prioritizing the active participation of partners, there is potential to foster a more supportive nutritional environment in the home and influence the nutrition of their offspring in the long term. Prenatal guidance must address not only the availability of healthy food but also its accessibility, along with culinary guidelines and cooking methods that prioritize ease of preparation and nutritional quality. Further research is needed to explore how to prioritize partner involvement in creating a supportive nutritional environment and investigate the broader roles of expectant fathers and non-male partners in shaping family lifestyles and long-term health outcomes and examine these dynamics among couples from different cultures.

## Figures and Tables

**Table 2 nutrients-16-04356-t002:** Food frequency during pregnancy and in the year following birth.

How Many Times Do You Eat the Following Foods Each Day? *8-Point Likert Scale (0–6+)*		During Pregnancy *n* = 13	Post Pregnancy *n* = 14
		Mean (SD)	Mean (SD)
***Desirable foods*** [33]	**Fruit**	1.2 (1)	1.1 (0.9)
	**Raw vegetables** (e.g., lettuce, tomatoes, and salad)	1.9 (1.4)	1.6 (1.0)
	**Cooked vegetables**, not including potatoes (e.g., carrots, courgettes, and broccoli)	1.3 (1.0)	1.1 (0.7)
	**No. of individuals consuming 5+ portions of fruit and veg per day**	6 (43%)	5 (36%)
***Less desirable foods*** [33]	**Crisps or savory snacks**	0.5 (0.8)	0.4 (0.3)
	**High-fat processed food** (e.g., cream, chips, and fried food)	0.4 (0.3)	0.4 (0.3)
	**High-sugar food** (e.g., sweets, cakes, cookies, and chocolates)	1.0 (0.8)	0.8 (0.5)
	**Processed meats** (e.g., hot dogs, burgers, and sausage)	0.4 (0.5)	0.3 (0.3)
	**Sweet beverages**	0.9 (1.9)	0.4 (0.6)
	**Alcoholic drinks**	0.3 (0.4)	0.5 (0.5)
	**Coffee/Black tea**	1.8 (1.3)	1.4 (0.9)
	**Low-calorie/Diet drinks**	0.5 (1.2)	0.3 (0.4)
**PHFE**	**Fruit accessibility ^a^**	11 (85%)	10 (72%)
	**Vegetable accessibility ^b^**	9 (69%)	6 (42%)

^a^ Number of participants who answered yes to “Without opening any opaque cupboard doors, is there any kind of fruit in your home now, displayed out in the open?”. ^b^ Number of participants who answered yes to “Do you have any ready-to-eat fresh vegetables on a shelf in the fridge or the kitchen counter now?”.

**Table 3 nutrients-16-04356-t003:** Theme descriptions, frequencies, and examples.

Theme		Partners n §		Example
		During Pregnancy *n* = 15	Post Pregnancy *n* = 12	
**Theme A**	**The importance of mutual prenatal PHFE decisions.** The significance of mutual decisions on food selection for the home prior to the birth.	14	11	*“Bringing food and drinks into the house is extremely important, especially when pregnant. Sometimes, she says let’s get a Coke. For me, that is a big “NO NO”. I don’t believe it is healthy…. I may be exaggerating, but we are no longer discussing it because we no longer buy it….. The decisions we made when we got married were strengthened when the desired pregnancy arrived.”* *(P.C; pregnancy).*
**Theme B**	**Increased motivation for a healthier PHFE during pregnancy.** Expectant fathers show greater motivation and interest during pregnancy than in fatherhood.	11	10	*“The first pregnancy is a time for taking care of someone, so it is important to be healthy… I may have less time, but I will be more motivated to make sure that my baby is healthy”* *(P.N; pregnancy).*
**Subtheme** **B.1.**	**Voicing a preference for a more enhanced environment than their own parents’ home.** Aspiring to create a healthier nutritional environment for their unborn child than they had.	8	8	*“I believe our house is healthier than the one I grew up in. My parents had more sweets, and I think we ate more processed food than we do now. We wish to create a healthier home”* *(P.A; pregnancy).*
**Theme C**	**Acknowledgement of their wives as ‘Nutrition Experts’.** Partners recognized their wives’ responsibilities and active role in nutritional decisions.	11	9	*“L. reads a lot. When we visited The Family Care Center or when she gave birth, the doctors were always shocked by the information she knew, she researches and explores everything…”* *(P.K; pregnancy).*
**Theme D**	**Challenges in accessibility of healthy foods in the home.** Highlighting a gap between the availability and accessibility of healthy foods, especially raw vegetables, cooked meals, and snacks.	12	11	*“I’ve learned that having healthy food at home is only half the battle. The real challenge is making sure it’s easy to grab and eat, especially when you’re tired and busy. Keeping fresh veggies and ready meals accessible can be really tough”. (P.O; father)*
**Theme E**	**“The Child Eats What We Eat”.** Fathers emphasized feeding their children what they themselves eat as parents, based on the household’s accessible food.	3	10	*“We as parents continued the same lifestyle we had before, and Ofir joined the same food environment we have at home since before birth”* *(P.G; father)*

## Data Availability

The original contributions of this study are presented within the article. For further information, inquiries may be directed to the corresponding author.

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
