# Peer review of "Exploring Partners’ Part in Shaping the Home Food Environment During the Transition to Fatherhood"

_nutrients, 2024, doi:10.3390/nu16244356_

Round 1

Reviewer 1 Report

Comments and Suggestions for Authors

This is an original study, but it has significant shortcomings which need to be remedied before it can be published. The following is a description of the main issues to be improved.

- The first time the Nestrition study is mentioned is in the objectives of the study, so it is not clear what the role of the Nestrition study is in this research. It is necessary that the theoretical framework of the introduction adequately explains what the Nestrition study is and what it consists of.

- It is not a longitudinal study, as there is no follow-up over time, but a simple qualitative pre-post study. In this case, it is a sequential study, but not a logitudinal one.

- The university or institution to which the ethical committee that approved the study belongs must be specified.

- The authors, year and validity of the Food Frequency Questionnaire (short FFQ) are unknown. Nor is it known why this instrument was used over others. If it was designed ad hoc it should be clearly specified.

- There are no inferential statistics of any kind, neither pre-post nor comparison with the other study (by the same authors). Although the number of participants is low, at least non-parametric inferential statistics should be applied.

- The qualitative analysis needs to clarify the scientific process followed for its application in the Method section more clearly.

Author Response

Exploring Partners' Part in Shaping the Home Food Environment During Transition to Fatherhood

RESPONSE TO THE REVIEWERS

REVIEWER 1

Thank you very much for taking the time to review this manuscript. Please find the detailed responses below and the corresponding revisions/corrections in green in the re-submitted manuscript file. Our point-by-point response is provided in green too in the table below.

This is an original study, but it has significant shortcomings which need to be remedied before it can be published. The following is a description of the main issues to be improved.

- The first time the Nestrition study is mentioned is in the objectives of the study, so it is not clear what the role of the Nestrition study is in this research. It is necessary that the theoretical framework of the introduction adequately explains what the Nestrition study is and what it consists of.

Thank you for this comment. This paper is part of the wider Nestrition study, which is fully described in the paper published in Nutrients in October 2024 (Peles, Chagit, Netalie Shloim, and Mary CJ Rudolf. "Nutritional Nesting (Nestrition): Shaping the Home Food Environment in the First Pregnancy." Nutrients 16.19 (2024): 3335). We agree that the Nestrition study should have been described more fully in the introduction in order to place this new aspect of the research in context. We have therefore provided some details about Nestrition in the introduction. [Lines 61-72]

- It is not a longitudinal study, as there is no follow-up over time, but a simple qualitative pre-post study. In this case, it is a sequential study, but not a longitudinal one.

We appreciate the reviewer's insightful comment. Upon reflection, we believe that the approximately 10-month interval between the two interviews justifies classifying our study as longitudinal. The term 'pre-post' often implies an intervention between time points, and although obviously significant, childbirth was not an intervention as such.

The literature supports defining qualitative studies with data collection at multiple time points as 'qualitative longitudinal research' (QLR). For instance, Lin (2023)1 describes QLR as a methodology used to capture critical moments, elements of experiences, behaviors, actions, or transitions in a continuous long-term project. Similarly, a study published in Sustainability conducted a meta-synthesis of qualitative longitudinal research to understand individuals' experiences of living with multimorbidity over time, exemplifying the application of QLR in health research2

In light of this, we have adopted the term 'two-phase longitudinal qualitative study' in our manuscript, as suggested by the second reviewer, and hope this terminology is acceptable. [Lines 16 and 79]

- The university or institution to which the ethical committee that approved the study belongs must be specified.

Thank you. The institution was in fact specified at the end of the paper in the section entitled ‘Institutional Review Board Statement’. For the sake of completeness we have added it to the methods section too [Line 163].

- The authors, year and validity of the Food Frequency Questionnaire (short FFQ) are unknown. Nor is it known why this instrument was used over others. If it was designed ad hoc it should be clearly specified.

We appreciate the reviewer highlighting this omission. The short FFQ used in our study was adapted from the Food Frequency Questionnaire validated by Hammond et al. (Hammond et al., 1993) and has been previously employed in our research on pediatric obesity(Willis et al., 2014). This adaptation was selected for its relevance to our study's focus and population. These details have now been clarified and referenced in the "Methods" section of the manuscript [line 116].

- There are no inferential statistics of any kind, neither pre-post nor comparison with the other study (by the same authors). Although the number of participants is low, at least non-parametric inferential statistics should be applied.

This is an interesting point, and a similar debate arose during the review process of the first paper where one of the reviewers requested statistical analysis on the sample of 15 women and the other felt it was inappropriate in view of the fact that the sample size was not powered to detect change. The quantitative data were collected primarily to describe participants’ sociodemographic and ‎nutritional characteristics, not for statistical analysis. ‎

The final decision for the earlier Nutrients paper was not to conduct statistical analysis, and the paper clarified the rationale for not applying inferential statistics in ‎the Methods section. The current paper states this [Line 159-160] in a similar way.

‘Given the sample size, inferential statistical testing was precluded when comparing measurements obtained at the two timepoints under study’.  

We believe that it is not appropriate to conduct comparative statistical analysis.

Please can we defer to the editor regarding this point?

- The qualitative analysis needs to clarify the scientific process followed for its application in the Method section more clearly.

We described the process of the qualitative analysis fully in the first paper (Peles, Chagit, Netalie Shloim, and Mary CJ Rudolf. "Nutritional Nesting (Nestrition): Shaping the Home Food Environment in the First Pregnancy." Nutrients 16.19 (2024): 3335.) but omitted to refer the reader to it. We have now provided that citation [line 128] and made some adjustments to the text which we believe further clarifies the scientific process employed.

Thank you again for your comments which we believe have significantly improved the quality and clarity of this paper.

1LIN, Pei-Ying. Qualitative Longitudinal Research. In: Varieties of Qualitative Research Methods: Selected Contextual Perspectives. Cham: Springer International Publishing, 2023. p. 401-406.)

2Maillot, Anne-Sophie, et al. "A Qualitative and Longitudinal Study on the Impact of Telework in Times of COVID-19." Sustainability 14.14 (2022): 8731.

Reviewer 2 Report

Comments and Suggestions for Authors

• The study examines how first-time pregnant women's male partners influence their physical home food environment (PHFE) as they adjust to fatherhood. It seeks to comprehend their nutrition-related knowledge, attitudes, and behaviors and how these may affect the family's eating patterns. According to the study, the first pregnancy is a crucial time for developing healthy PHFE. Individual interviews were conducted after completing an online questionnaire as part of a two-phase longitudinal qualitative study. The study evaluated the lifestyle, nutritional practices, and interest in recommendations for a healthier PHFE of fifteen male partners of primigravida women. The results show that prospective fathers are very motivated to give their kids a healthier environment than they had as kids.

The study is a quantitative study carried out with male partners and fathers of primiparous women who have undergone nesting nutrition training (NESTRITION). The concept of nurturing is fundamental because it develops the importance of integrality for quality of life and nutritional support, with a positivist approach and support from the partner and the community. In a consensus published in the journal Appetite, Jess Haines and collaborators (doi: 10.1016/j.appet.2019.02.007) established a common point of view on these aspects, which in the present work refers to the period of preparation during the gestational period, up to the phase of introduction of complementary feeding.

Another exciting aspect of neonatal nutrition is the period of preparation for pregnancy when women stop contraceptive methods and start trying to conceive. This average period of 3 months led to the concept of a 12-month gestation, which was later expanded to the period of exclusive breastfeeding, with the idea of an 18-month gestation. Hence, the concept of 1000 days and 2200 days of development was established as landmarks for growth, immunity, and neural development.

One of the criticisms we can analyze of the work of Chagit Peles and her team is that by evaluating only male individuals, they fail to analyze different partners who are not defined as male. Obviously, one of the most complex limitations of the study is that, in addition to the small number for the quantitative area of the project, they are all from the same age group. These young people are older than most parents of primiparas in other countries. Most of them have an excellent level of education and can be from a higher social class. There needs to be a description of the criteria for analysis in this respect, as they responded to calls from social projects, the media, hospitals, and the press. The motivation of these individuals may be greater than that of those who didn't apply.  The reality in Israel is different from other countries, as 18-year-olds spend three years in the army, with all the characteristics of physical and mental maturation, with traumas or gains. In some way, this has to do with their knowledge, preparation, and experience of feeding themselves and their families. And we're not talking here about female conscription, which is also compulsory.

Male parents may not be new to pregnancy. There is no description of whether any of the participants have already had other relationships with pregnancy and experiences with maternal and infant feeding.

Another critical factor that needs to be described in the project is that the study was carried out precisely during the first two years of the pandemic. There is no analysis of the changes in eating habits during this period, including the lockdown, the reduction in visits to health services, the concept of pandemic babies, the use of delivery services, the greater awareness of shared eating, meal preparation, buying locally, and other processes.

Moving on to the project itself, there are a few things to analyze: the concept of household food insecurity, based on lifestyle and health feelings, is at odds with the worldwide concept of individual or family food insecurity, based on access to at least one meal a day.

With regard to the results, we want to understand whether there was an improvement in the quality of life of the individuals based only on the slight increase in the consumption of fruit and vegetables. Couldn't the slight increase in sweets and snacks be aimed at guests, as one of the participants pointed out?

Regarding the qualitative aspects, the decision to involve the partner in defining the diet is indeed exciting. There is no description of this issue since the decision to start, keep, or abandon breastfeeding is both ways. There needs to be motivation, planning, and implementation of exclusive breastfeeding (how many started, how many weaned, and at what age). Partners can once again be great encouragers or saboteurs at the first sign of a crisis, such as babies crying continuously. The fact that they are mothers who have been through previous childcare with guidance and education dramatically modifies the feeding prognosis.

Finally, a thematic suggestion: who is responsible in the partnership for planning, organizing, and preparing the shopping? The parents answered that the mother is usually the nutrition expert. Does this also apply to shopping? Could it be that the increase in practical food is not an experience for the parents, thinking that the mother will not always be able to cook the food during pregnancy, the first few months, and afterward?

Author Response

Exploring Partners' Part in Shaping the Home Food Environment During Transition to Fatherhood

RESPONSE TO THE REVIEWERS

REVIEWER 2

Thank you very much for taking the time to review this manuscript. Please find the detailed responses below and the corresponding corrections in green in the re-submitted manuscript file. Our point-by-point response is provided in green too in the table below.

Comments and Suggestions for Authors

The study examines how first-time pregnant women's male partners influence their physical home food environment (PHFE) as they adjust to fatherhood. It seeks to comprehend their nutrition-related knowledge, attitudes, and behaviors and how these may affect the family's eating patterns. According to the study, the first pregnancy is a crucial time for developing healthy PHFE. Individual interviews were conducted after completing an online questionnaire as part of a two-phase longitudinal qualitative study.

The study evaluated the lifestyle, nutritional practices, and interest in recommendations for a healthier PHFE of fifteen male partners of primigravida women. The results show that prospective fathers are very motivated to give their kids a healthier environment than they had as kids.

The study is a quantitative study (we presume this is a typographical error as it is a qualitative study) carried out with male partners and fathers of primiparous women who have undergone nesting nutrition training (NESTRITION). The concept of nurturing is fundamental because it develops the importance of integrality for quality of life and nutritional support, with a positive approach and support from the partner and the community. In a consensus published in the journal Appetite, Jess Haines and collaborators (doi: 10.1016/j.appet.2019.02.007) established a common point of view on these aspects, which in the present work refers to the period of preparation during the gestational period, up to the phase of introduction of complementary feeding.

Another exciting aspect of neonatal nutrition is the period of preparation for pregnancy when women stop contraceptive methods and start trying to conceive. This average period of 3 months led to the concept of a 12-month gestation, which was later expanded to the period of exclusive breastfeeding, with the idea of an 18-month gestation. Hence, the concept of 1000 days and 2200 days of development was established as landmarks for growth, immunity, and neural development.

Thank you for highlighting the importance of the perinatal period and for sharing new ideas that are emerging in this field. We would like to clarify one point regarding our study. The research was conducted on primiparous women and their partners; however, they had not undergone training in nesting nutrition. To our knowledge, such training has not yet been implemented in any setting. The participants were de novo, and the study was designed to explore their knowledge, attitudes, and practices during the critical period of first pregnancy and the transition to parenthood. Specifically, the study aimed to assess their perspectives on whether the first pregnancy provides a unique opportunity to foster a healthy home food environment.

This research was undertaken with the hope that it could serve as a foundational step toward implementing structured training and guidance for prospective parents in the future.

- One of the criticisms we can analyze of the work of Chagit Peles and her team is that by evaluating only male individuals, they fail to analyze different partners who are not defined as male. Obviously, one of the most complex limitations of the study is that, in addition to the small number for the quantitative area of the project, they are all from the same age group. These young people are older than most parents of primiparas in other countries. Most of them have an excellent level of education and can be from a higher social class. There needs to be a description of the criteria for analysis in this respect, as they responded to calls from social projects, the media, hospitals, and the press, of these individuals may be greater than that of those who didn't apply.  The reality in Israel is different from other countries, as 18-year-olds spend three years in the army, with all the characteristics of physical and mental maturation, with traumas or gains. In some way, this has to do with their knowledge, preparation, and experience of feeding themselves and their families. And we're not talking here about female conscription, which is also compulsory.

We thank the reviewer for the thoughtful and detailed comments. We would like to clarify that many of the points raised were already addressed. Specifically, the limitations related to participant demographics, including gender, age, and education, were discussed in the final paragraph of the discussion section [499-505]. The point regarding participant motivation is interesting and has now been explicitly added [lines 499–500].

Regarding the inclusion of non-male partners:  We appreciate the reviewer’s point about the need to include diverse family structures. This was indeed a consideration in the design of our study, and is reflected in our suggestions for further research in the conclusions [line 525]:

 "Further research is needed to explore how to prioritize partner involvement in creating a supportive nutritional environment and investigate the broader roles of expectant fathers and non-male partners in shaping family lifestyles and long-term health outcomes and examine these dynamics among couple from different cultures." 

Given that this is a preliminary study in an under-researched field, we specifically focused on couples rather than individuals to identify gaps and dynamics that could serve as a robust foundation for developing future evidence-based interventions. Our study aimed to include couples in their first pregnancy without specifying the gender of the partner. Recruitment was open to all couples within the general population in Israel. However, all the couples who responded to our recruitment efforts were heteronormative. This reflects the demographics of those who chose to participate, rather than a deliberate focus on heteronormative couples. 

Given that this was a preliminary study in an under-researched area, we focused on the sample that emerged from our recruitment efforts. This initial work therefore provides a foundational understanding that can inform future research and interventions, which we hope will include a broader range of family structures.

Regarding the generalizability of the findings: As noted in the original manuscript, the relatively homogeneous demographics of our sample—age, education level, and socioeconomic status—pose limitations to the generalizability of our findings. We chose to focus on first-time pregnancy among couples in one country to create a baseline understanding in this novel research area. While this approach has clear limitations, it allowed us to generate focused insights that can be expanded upon in the future coming studies. The unique cultural context was also acknowledged in the manuscript (line 501). These factors highlight the importance of studying diverse cultural settings in future work. 

Expanding research scope: We agree with the reviewer’s suggestion that future research should include diverse cultural contexts and populations. As part of the broader Nestrition research program, we have already extended our work to the UK, with two additional papers currently in progress comparing findings from first-time pregnant women and their partners in both Israel and the UK. 

To address the reviewer’s comment, we have revised the conclusions to emphasize the importance of studying couples in varied cultural settings: 

"Further research is needed to explore how to prioritize partner involvement in creating a supportive nutritional environment, investigate the broader roles of expectant fathers and non-male partners in shaping family lifestyles and long-term health outcomes and examine these dynamics among couples from different cultures." [line 534] 

We hope these clarifications and revisions address the reviewer’s concerns while highlighting the thoughtfulness and scope of our original manuscript.

- Male parents may not be new to pregnancy. There is no description of whether any of the participants have already had other relationships with pregnancy and experiences with maternal and infant feeding.

Thank you. This is an important omission. The interviews did probe into potential other pregnancies that the men may have fathered. All of them were ‘de novo’ and this is now added to the results [Line 169]

-Another critical factor that needs to be described in the project is that the study was carried out precisely during the first two years of the pandemic. There is no analysis of the changes in eating habits during this period, including the lockdown, the reduction in visits to health services, the concept of pandemic babies, the use of delivery services, the greater awareness of shared eating, meal preparation, buying locally, and other processes.

While the primary focus of our study was on the Physical Home Food Environment and the transition to parenthood, we did consider the unique context in which the research was conducted. Interviews were primarily conducted during late 2020 till the end of 2021, a time when widespread vaccinations were already underway in Israel, and participants were not under lockdown. These contextual details are now incorporated into our discussions section to explicitly reflect this context and its potential effects on participants’ responses [line 505-508]. but we appreciate the opportunity to expand on this further.

In the follow-up questionnaire, specific questions were included to gauge the perceived impact of the pandemic. Three fathers reported that the pandemic had had little effect on what the baby ate and the rest responded it made no difference at all, with similar results on the effect on their own diet.

We hope this clarification and the additional detail address the reviewer’s concerns and enhance the contextualization of our findings.

Moving on to the project itself, there are a few things to analyze: the concept of household food insecurity, based on lifestyle and health feelings, is at odds with the worldwide concept of individual or family food insecurity, based on access to at least one meal a day.

We would like to clarify that food insecurity in our study was assessed using two self-reported measures adapted from Radimer et al. (2006) : (1) "The food that we bought just didn’t last, and we didn’t have money to get more,” and (2) "We worried whether our food would run out before we got money to buy more" (as stated in lines 119-121). These measures align with established approaches for evaluating household food insecurity in industrialized countries and have also been adopted by the USDA as part of their Household Food Security Survey Module.

The responses from our sample indicated that participants were not experiencing food insecurity. This was noted in the limitation section (line 503), where we highlighted that the findings reflect a population with no evidence of food insecurity. ‎

-With regard to the results, we want to understand whether there was an improvement in the quality of life of the individuals based only on the slight increase in the consumption of fruit and vegetables. Couldn't the slight increase in sweets and snacks be aimed at guests, as one of the participants pointed out?

Assessing improvement in quality of life was beyond the scope of our study, which primarily focused on the Physical Home Food Environment. However, we did include participants' self-reported Health Satisfaction scores, based on the WHO Assessment tool1 as shown in Table 1 and referenced in the manuscript (line 182). Given the necessarily small sample size—characteristic of a study with a predominantly qualitative focus—we were cautious not to draw definitive conclusions about individual changes based on the Food Frequency Questionnaire (FFQ) data. While the slight increase in the consumption of fruits and vegetables is noteworthy, the modest rise in sweets and snacks could indeed reflect external factors, such as offering snacks to guests, as one participant noted during the interviews. This context is discussed qualitatively within the results section to provide a more nuanced understanding.

‎ -Regarding the qualitative aspects, the decision to involve the partner in defining the diet is indeed exciting. ‎There is no description of this issue since the decision to start, keep, or abandon breastfeeding is both ways. ‎There needs to be motivation, planning, and implementation of exclusive breastfeeding (how many started, ‎how many weaned, and at what age). Partners can once again be great encouragers or saboteurs at the first ‎sign of a crisis, such as babies crying continuously. The fact that they are mothers who have been through ‎previous childcare with guidance and education dramatically modifies the feeding prognosis.‎

Our study focused exclusively on the Physical Home Food Environment and did not address aspects related to breastfeeding.

While breastfeeding is undoubtedly an important topic closely tied to family health, it lies outside the scope of our research. Our aim was to explore the dynamics and involvement of the partner in shaping the PHFE, including food selection, procurement, and storage.

We are somewhat concerned that the reviewer (as seen in a previous comment) believed that ‎Nestrition is an intervention, which we hope we have made clear that it is not. Certainly, the ‎mothers were likely to have had routine guidance through the universal national Mother ‎and Infant clinic framework in Israel but as was clear in the results we obtained in both this ‎paper and the earlier Nestrition paper, there is no focus on the PHFE.‎

Finally, a thematic suggestion: who is responsible in the partnership for planning, organizing, and preparing the shopping? The parents answered that the mother is usually the nutrition expert. Does this also apply to shopping? Could it be that the increase in practical food is not an experience for the parents, thinking that the mother will not always be able to cook the food during pregnancy, the first few months, and afterward?

Shopping emerges as a key aspect in two of the main themes identified in the research (see Themes C and D, [lines 319 to 346]), where the dynamics of food acquisition and preparation are discussed in detail. These themes are further elaborated in the ‘Results’ and ‘Discussion’ sections, particularly in relation to the relationship between role divisions, perceptions of expertise, and the evolution of food practices during the transition to parenthood (see lines 449 to 455, for example).

We were uncertain about the reviewer’s intended meaning regarding the "increase in practical food." If this refers specifically to convenience foods, we have addressed their role in the context of the participants’ reported behaviors. If additional clarification is needed, we would be happy to expand on this further.

Thank you again for your comments which were challenging, but we believe have significantly improved the quality and clarity of this paper.

  1. Skevington SM, Lotfy M, O’Connell KA. The World Health Organization’s WHOQOL-BREF quality ‎of life assessment: Psychometric properties and results of the international field trial a Report from the ‎WHOQOL Group. Qual Life Res. 2004;13(2):299-310. ‎doi:10.1023/B:QURE.0000018486.91360.00/METRICS),

Round 2

Reviewer 1 Report

Comments and Suggestions for Authors

In my view, the authors have improved their work adequately to proceed with the publication of the manuscript. I send my congratulations to the authors.

Reviewer 2 Report

Comments and Suggestions for Authors

After reading all the clarifications from the authors, I believe that the majority of my questioning issues were adequately treated, and I do not have any more concerns.